
**Analysis of Trade-offs between Food Security and Water-Land**
**Savings through Food Trade and Structural Changes of Virtual**
**Water Trade in the Arab World**
Sang-Hyun Lee[1], Rabi H. Mohtar[1], Seung-Hwan Yoo [2]
[1]Department of Biological and Agricultural Engineering, Texas A&M University, College station, TX77840, USA
[2]Department of Rural and Bio-systems Engineering, Chonnam National University, Gwangju, Republic of Korea
*Correspondence to*: Rabi H. Mohtar (mohtar@tamu.edu)
**Abstract**
The aim of this study is to analyze the impacts of food trade on food security and water-land savings in the Arab World in
terms of virtual water trade (VWT). We estimated the total volume of virtual water imported for four major crops—barley,
maize, rice, and wheat—from 2000 to 2012, and assessed their impacts on water and land savings, and food security. The
largest volume of virtual water was imported by Egypt (19.9 billion m³/year), followed by Saudi Arabia (13.0 billion m³/
year). Accordingly, Egypt would save 13.1 billion m³ in irrigation water and 2.1 million ha of crop area through importing
crops. In addition, connectivity and influence of each country in the VWT network was analyzed using degree and
eigenvector centralities. The study revealed that the Arab World focused more on increasing the volume of virtual water
imported during the period 2006-2012 with little attention to the expansion of connections with country exporters, which is a
vulnerable expansion. This study shed light on opportunities and risks associated with VWT and its role in food security and
land management in the Arab World.
Keyword: Food security, Arab World, Virtual water trade; Degree centrality; Eigenvector centrality
**1 Introduction**
Food trade is an important element of food security in water-scarce regions (Konar et al., 2012; Hanjra and Qureshi, 2010;
Hoekstra, 2003) because food trade drives water conservation or loss in terms of the virtual water trade (VWT), which refers
to the trade of water embedded in food products (Allan, 1993; Aldaya et al., 2010; Antonelli and Tamea, 2015). The concept
and quantitative estimates of virtual water can help in realistically assessing water scarcity for each country, projecting
future water demand for food supply, increasing public awareness about water, and identifying water-wasting processes in
production (Oki and Kanae, 2004). For water-scarce countries, achieving water security through importing water intensive
products could be a more attractive option, compared to producing all water-demanding products domestically (Hoekstra and
Hung, 2005). The global volume of international crop-related virtual water flows averaged 695 billion m³/year over the
period 1995–1999, meaning that 13% of the water used for crop production in the world was not used for domestic
consumption but rather for export in virtual form (Hoekstra and Hung, 2005). The International Water Management Institute
(IWMI) and the Government Office for Science both state that the VWT could contribute to relieve water stress through
using global water more efficiently, in the event of an increase in global food trade (Government Office for Science,
London, 2011; Molden, 2007). In addition, Falkenmark and Lannerstad (2010) estimated that it would be necessary to
double the VWT by 2050 to compensate for agricultural water deficits.
The VWT has been suggested as relevant to the water policy of a nation (Schyns and Hoekstra, 2014), providing a new point
of view from which both food security and sustainable water management are considered (Novo et al., 2009). The VWT and
the respective savings garnered through the trade of agricultural goods have been quantified in a number of studies. Oki and



Kanae (2004) investigated whether VWT could save global water resources and determined that "real water" in exporting
countries tends to be smaller than "virtual water" in importing countries. For example, approximately 1140 km³/year of
virtual water was imported through the food trade, e,g., cereals, soybeans, and meat; however, 680 km³/year of real water
was used to produce those foods in exporting areas. This is due to the difference between crop water requirement between
the importing and exporting country, with the later usually lower. Yang et al. (2006) revealed that the VWT could generate a
global water saving because virtual water has flown primarily from countries of high crop water productivity to countries of
low crop water productivity. In their study, globally, 336.8 km³/year of virtual water was saved by the international trade of
major food crops from 1997 to 2001, and 20.4% of the total global net virtual water import was imported to countries that
have water availability below 1700 m³ per capita, such as Arab countries. Fader et al. (2011) showed that the trade of crop
products saves 263 km³/year of virtual water, globally, representing 3.5% of the annual precipitation on cropland. In
particular, water-scarce countries, such as China and Mexico, but also The Netherlands and Japan, saved large amounts of
water by importing goods—from 25 to 73 km³ of water—because they would need relatively large amounts of water to
produce the goods they import. According to the study by Biewald et al. (2014), blue water saving from international trade
can bring enormous benefits in water-scarce regions; for example, 17 billion m³ of blue water per year were saved by the
global food trade, and the value of blue water saving was estimated to 2.4 billion US$.
Previous studies showed that the effective import of virtual water may reduce water use for domestic food production in
importing countries and help alleviate water stress in water-scarce regions, such as the Arab World where the largest water
deficit in the world exists (Gleick, 2000; World Bank, 2009). The critical condition of water scarcity in the Arab world will
reach severe levels by 2025 (Tolba, 2009). In addition, if population increases rapidly and urbanization continues fast,
availability of water could be reduced in Arab countries by about 50% by the year 2025 (Abahussain et al., 2002). Water
shortages will certainly speed up the rate of desertification in the Arab countries with a larger deficit in freshwater
(Abahussain et al., 2002). Agricultural water withdrawals account for over 85% of the total water withdrawn throughout the
many countries of the Arab World (FAO, 2014). Irrigation systems in the Arab World are based on pumping groundwater
resources such as aquifers, and water security is being threatened by declining aquifer levels and the extraction of non-
renewable groundwater (Antonelli and Tamea, 2015). In addition, Immerzeel et al. (2011) expected that the unfulfilled water
demand in the entire Arab World would increase from the current level of 16% to 51% in 2040–2050 due to climate change.
The IPCC projections also indicate that rainfall in the Arab region will become intense, and dry spells will become more
pronounced. In addition, the zone of severely-reduced rainfall extends throughout the Mediterranean region and the northern
Sahara (Hennessy et al., 2007). Milly et al. (2005) identify that climate change causes a drop in water run-off by 20% to 30%
in most of Middle East North Africa (MENA) by 2050, mainly due to rising temperatures and lower precipitation. In
addition, the regions including Syria, Lebanon, Israel, and Jordan will get drier, with significant rainfall decrease in the wet
season.
Accordingly, food trade can be regarded as the most important factor for saving domestic water resources and decreasing
water stress in addition to improving food security in the Arab World. This study addresses three questions that relate to the
role and impact of the VWT in the Arab World, which are raised to draw attention to the complexity of the issue and the
need for a broader view in assessment. These questions are: 1) What are the effects of the VWT on water savings and land
tenure in the Arab World, 2) Has the structure of the virtual water import in the Arab World been vulnerable or robust? 3)
Who are the influential importers and exporters in VWT network in the Arab World?
The aim of this study is to analyze the quantitative and structural characteristics of VWT in the Arab World in order to
understand the effects on water savings and land tenure from importing crops and identify the temporal changes of VWT
structure.
First, we estimated the total volume of virtual water imported through four major crops—barley, maize, rice, and wheat—in
the Arab World from 2000 to 2012, and the effects of importing crops on water and land savings were evaluated in each

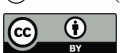



Arab country. However, food import can cause a decrease in local food production, which can be particularly a critical issue
in the Arab World. Accordingly, we estimated water requirement of increasing 1 % self-sufficiency of study crops in
comparison to average self-sufficiency from 2000 to 2012 in terms of trade-off between water saving and food self-
sufficiency.
Second, we analyzed the structural characteristics of the VWT in the Arab World using degree centrality, which represents
the connectivity of a node in a network system, and categorized the countries in the Arab World in terms of vulnerable
expansion (or reduction) and robust expansion (or reduction) in the VWT network. In addition, influence of each country
was analyzed using eigenvector centrality to identify influential countries who could affect the entire VWT network in the
Arab World. Understanding the VWT structure is important as quantifying the amount of import and export. Recent
literature has emphasized the change in structure of the VWT in terms of a network approach (Dalin et al., 2012; Konar et
al., 2012; Lee et al., 2016).

## 2 Materials and Methods

### 2.1 Calculation of a virtual water trade using food trade and water footprint

The VWT represents the water embedded in international trade, and the main factors for quantifying a VWT are trade data
and water footprint (WFP, m³/ton), which is the volume of water used for producing one ton of crops. Therefore, a VWT is
calculated by multiplying the trade by its associated water footprint, as follows:
$$VWT\ [n_e, n_i, c, t] = CT\ [n_e, n_i, c, t] \times WFP\ [n_e, c], \qquad (1)$$
in which variable VWT denotes the VWT from the exporting country, ne, to the importing country, ni, in year t, as a result
of trade in crop c; CT represents the crop trade from the exporting country, ne, to the importing country, ni, in year t as a
result of trade in crop c; and WFP represents the water footprint of crop c in the exporting country, ne.
The WFP of a crop is derived from the crop water requirement (m³/ha) per yield (kg/ha), as follows:

$$WFP[c] = \frac{CWR[c]}{Production\ [c]}, \qquad (2)$$

where WFP (m³/ton) is water footprint of a crop c, CWR is the crop water requirement, and Production is the yield per year.
The water footprint for a crop is divided into green and blue water footprints, based on the water resources (Hoekstra and
Chapagain, 2008). Green water footprint indicates that water supplied by precipitation is retained in the soil of the root zone
(Falkenmark, 1995), and blue water footprint is the water stored at the surface or in the ground. Therefore, green water
footprint is related to rain-fed agriculture and blue water footprint is related to irrigation water provided by aquifers or
surface bodies of water.

### 2.2 Quantification of water and land savings by importing crops using water footprint and land productivity

The import of crops could affect the water and land savings in the importing country. Therefore, the failure of trade could
cause water and land shortages in the Arab World. Therefore, we analyzed water and lands requirements for producing as
much crop as is imported in each Arab country. In other words, the water and land savings indicated resource requirements
needed by the shift from crop import to domestic production. Although this assumption about water and land savings
considers an extreme trade situation, these results could be used to understand how the international crop trade is important
in the Arab World in terms of water and land savings. The national water and land savings indicated the amount of blue
water and land requirements for substituting crops imported to domestic production. Thus, it was calculated as follows:
$$Water\ saving_{c,i} = Import_{c,i} \times Blue\ water\ footprint_{c,i} \qquad (3)$$
$$Lands\ saving_{c,i} = Import_{c,i} \times \frac{Lands_{c,i}}{Production_{c,i}} \qquad (4)$$

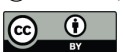



where c and i indicate crop and importer, and w indicates the water resource such as ground water, surface water, and treated
water.

**2.3 Analysis of degree and eigenvector centrality in the virtual water trade network**

The VWT network indicates flows of virtual water among countries through crop trade, and thus, it consists of volume and
links. In this study, we considered both volume and links of the VWT network for identifying changes in VWT structure, for
example, vulnerable expansion (or reduction) and robust expansion (or reduction). Therefore, it is important to estimate the
connectivity in a VWT network. Therefore, we applied the degree centrality, which is the number of edges incident on a
given node (Freeman 1979). Degree centrality is divided into in- and out-degree centralities, depending on the direction, and
the in-degree centrality of each Arab country was calculated because we focused more on the import of virtual water in the
Arab World. An importer accompanying a high in-degree centrality has expanded connectivity with exporters, meaning that
this importer could cope with an accidental disconnection from a certain exporter. A few studies that analyze the structure of
the VWT using a network-based approach have been conducted (Konar et al., 2012; Dalin et al., 2012; Lee et al., 2016). The
degree centrality of the VWT is:
$C_i = \sum_j^N VWT_{ij}/(N-1),$                    (5)
where $C_i$ is the degree centrality of country i and N is the number of total countries. $VWT_{ij}$ is the link between the ith and
jth countries.
The entire network can be affected by a few nodes, which is influential nodes, and it is important to identify these nodes for
understanding and estimating the change of entire network system. An eigenvector centrality can measure important and
influence of each node in the entire network, and it is related not only of own connection but also connection of other node
which connects to own. Therefore, a node is more influential if it is in relation with the nodes that are, themselves, influential
(Ruhnau, 2000). The eigenvector centrality assigns relative centrality to all of the nodes in the network, based on the
principle that connections to high-level centrality nodes contribute more to the centrality of the nodes than equal connections
to low-level centrality nodes (Ruhnau, 2000; Lee et al., 2016). Therefore, the eigenvector centrality of node is related to both
the number of links to partners and their centrality (Ruhnau, 2000). Bonacich (1972) defined the centrality $c(v_i)$ of a node $v_i$
as the positive multiple of the sum of adjacent centralities, as follows:
$\lambda c(v_i) = \sum_{j=1}^n \alpha_{ij} c(v_j) \qquad \forall i.$                    (6)
In matrix notation, with $c = (c(v_i), \dots, c(v_n))$, the above equation yields
$Ac = \lambda c$                    (7)
Eigenvector centrality is determined by calculating the principal eigenvector that has the largest eigenvalue among every
eigenvector. An eigenvector of the maximal eigenvalue with only non-negative entries does exist, and we call a non-negative
eigenvector ($c \geq 0$) of the maximal eigenvalue the principal eigenvector, and we call the entry $c(v_i)$ the eigenvector-
centrality of node $v_i$ (Ruhnau, 2000).

**2.4 Data collection and limitations of data availability**

A main data set was international trade, and the international trade data of the study crops from 2000 to 2012 was obtained
from FAOSTAT (http://www.fao.org/faostat/), as shown in Table 1. The crop with the largest amount of import was wheat,
with 359.7 million ton imported by the Arab World from 2000 to 2012, followed by maize (187.2 million ton), barley (116.4
million ton), and rice (49.0 million ton). Most of the Arab countries increased the imports of the four major crops from 2000
to 2012. In particular, the largest increase was represented in Egypt, for example, the amount of the imported crops in Egypt
was 11.2 million ton in 2000 and it increased to 18.0 million ton in 2012.
To quantify VWT and assess its effect on water and land savings, water footprint data of crops was essential. However,
water footprint of crops is based on crop water requirement and irrigation, thus various data are required for calculating it,

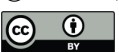



for example climate data, crop information, irrigation scheduling, and soil characteristics. In addition, each variable is
dependent on local characteristics, thus the study for national water footprint should be executed for each country, basin, or
specific area, and it was out of the scope of this study. Therefore, the estimation of water footprint was not included but we
applied water footprint data set from the study executed by Mekonnen and Hoekstra (2010). They estimated the average
value of green and blue water footprints of crops and crop products at the national level from 1996 to 2005. In addition, the
blue water footprint and land productivity for each country in the Arab World were applied to assess effects on water and
land savings from importing crops. The blue water footprint for each country in the Arab World was also obtained from
Mekonnen and Hoekstra (2010). Land productivity was calculated by the harvest area and crop production, which were
collected from FAOSTAT (http://www.fao.org/faostat/), as shown in Table 2. Internal water resource and land area in each
country were collected from World Bank (http://data.worldbank.org).
However, time scales of international trade were different from water footprint data. For example, water footprints used in
this study were based on data from 1995 to 2005; however, we applied the food trade data from 2000 to 2012. Therefore, the
application of average water footprint to time-series trade data can cause a false estimate of the effects of VWT. However,
the water footprint data indicated the representative index using average value, and the part of periods for water footprint is
overlapped with the period of trade data. Therefore, even if there is limitation of data availability, the water footprint data
from Mekonnen and Hoekstra (2010) can be used for estimating VWT in this study.
**Table 1.** The amount of crops imported by the Arab World from 2000 to 2012 (FAOSTAT).
**Table 2.** Cultivation area and production of four major crops in the Arab World.
**3 Results and Discussion**
**3.1 Quantification of virtual water trade in the Arab World from 2000 to 2012**
The total amount of green and blue water imported by each Arab country from 2000 to 2012 reached 921.2 and 80.5 billion
m³, respectively, in the Arab World, is shown in Table 3 and Figure 1. The largest volume of green water was annually
imported by Egypt (19.1 billion m³/year), followed by Saudi Arabia (11.9 billion m³/year). In addition, the largest amount of
blue water was imported annually by Saudi Arabia (1.2 billion m³/year), followed by the UAE (0.9 billion m³/year). Over 70%
of the green water imported into the Arab World annually through the barley trade (approximately 8.5 billion m³/year) went
to Saudi Arabia. The amount of virtual water imported through the trade of maize was 13.0 billion m³/year, with Egypt as the
primary importer, importing 31% of the total imported into the Arab World. Rice is a blue-water-intensive crop, and the
importers of rice also import a lot of water. About 3.0 billion m³/year of blue water were imported in the rice trade from 2000
to 2012, and Saudi Arabia, the UAE, and Iraq were the primary importers. The largest volume of virtual water imported by
the Arab World was due to wheat trade. The annual amount of virtual water imported through wheat trade in the Arab World
from 2000 to 2012 was approximately 42.6 billion m³/year, but the amount of blue water was only 2.0 billion m³/year. Over
35% of the virtual water imported through the wheat trade was imported by Egypt (15.7 billion m³/year).
The volume of virtual water imports per capita (VWIcap) indicates how the countries are dependent on water resources from
abroad. Figure 2 shows that the VWIcap was 1266.6 m³/cap/year in the UAE, which was the largest value in the Arab World.
The UAE is strongly dependent on the import of virtual water, even though the UAE imports only 4.2 billion m³/year of virtual
water. The VWIcap increased significantly in Saudi Arabia and Libya from 2000 to 2012. Saudi Arabia and Libya imported
about 453.4 and 497.8 m³/cap/year, respectively, of virtual water more in 2012 than in 2000. Saudi Arabia was the second
biggest importer in the Arab World, and its VWIcap was also the fifth highest in the Arab World. In the condition of increasing
population, the VWIcap in the Arab World can be used to estimate the requirement of virtual water import in future, and it
contribute to set water and food management for increasing domestic production and decreasing the VWIcap in the Arab
World.



We also focused on the volume of virtual water exported to the Arab World by each exporter from 2000 to 2012 (Figure 3).
Through barley trade, Ukraine exported 41.1 billion m³ of green water to the Arab World, making up 27% of the total green
water imported in the Arab World through barley. In terms of blue water traded through barley, five exporters (Germany,
Australia, the Russian Federation, Ukraine, and India) provided 78% of the total blue water imported in the Arab World
through barley. In the VWT via maize, Argentina contributed 40% of the total amount of green water imported by the Arab
World through maize, but the blue water imported by the Arab World was primarily from the USA.  In the VWT via rice, the
major virtual water exporters to the Arab World were India, Thailand, and Pakistan. In particular, 30.4 billion m³ of blue water
was imported from these countries from 2000 to 2012, which comprised 78% of the blue water imported by the Arab World
through rice. Wheat was the most representative crop imported by the Arab World. The Russian Federation and the USA
provided 25% (140.6 billion m³) and 21% (111.2 billion m³), respectively, from 2000 to 2012, of the total amount of green
water imported in the Arab World through wheat, and the remaining 55% was divided among several exporters, including
Australia, Canada, France, and Ukraine.
**Table 3.** The amount of virtual water imported by the Arab World from 2000 to 2012.
**Figure 1.** The total amount of virtual water imported by each country in the Arab World from 2000 to 2012, separated into
green (upper) and blue (lower) water. The pie graph shows the annual import and proportion of each crop, and the size of the
pie indicates the amount of annual virtual water imported from 2000 to 2012.
**Figure 2.** Virtual water imports per capita in 2000 and 2012.
**Figure 3.** The amounts of green water export (GWE) and blue water export (BWE) from the primary exporters to the Arab
World from 2000 to 2012.

**3.2 Assessment of trade-offs between food self-sufficiency and water-lands savings through food trade in the Arab World**

Crop import could result in low food self-sufficiency in the Arab World, but water and land savings benefits of VWT. This
study shows which countries were more successful in achieving water or land savings through importing crops. The national
resource managers and trade policy makers in the Arab World would benefit from better understanding of the relationship
between international trade and the preservation of national resources, and these results could provide useful information to
each country in the Arab World.
Table 4 shows that water saving by crop import in Saudi Arabia was 8.14 billion m³/year, 3.4 times larger than its internal
water resources (2.40 billion m³). However, the land saving was 1.5 million ha, making up 0.9% of the total agricultural lands
in Saudi Arabia, which indicates that the crop trade in Saudi Arabia has more significant benefit in terms of water resource
than land resource. Egypt and the UAE were also strongly influenced by the impact of crop import on water saving. On the
other hand, Lebanon saved 0.06 billion m³ of water resources annually through crop import, which was only 1.3% of its internal
water resources. However, the crop import could bring a large amount of land saving; for example, about 0.24 million ha could
be saved by crop import, comprising over 30% of the agricultural area in Lebanon. In addition, in Jordan and Kuwait, crop
imports could have a strong impact on land saving.
However, increasing food imports is also correlated to decreasing domestic food production. Accordingly, it is important to
understand the trade-off between water saving and food self-sufficiency in the Arab World. In this study, we defined self-
sufficiency of crops as the ratio of imported crops to total consumption, and estimated the amount of blue water footprint for
increasing self-sufficiency of crops by 1% in comparison to average self-sufficiency from 2000 to 2012, as shown in Table 5.
For example, the average self-sufficiency of wheat in Egypt from 2000 to 2012 was 47.64 % and 278.77 million m³ irrigation
water would be required to increase self-sufficiency by 1%, in order to reach 48.64 %. The self-sufficiency of wheat in Saudi
Arabia was 74.02 % and 118.11 million m³ for increasing self-sufficiency by 1%. In contrast, the self-sufficiency of wheat in
Tunisia was 46.05 % but the water requirement for increasing self-sufficiency by 1% was only 3.84 million m³. As shown in

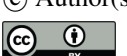



results, increase of food security accompanies a lot of water requirement in the Arab World and these results can give the
useful information for analyzing trade-off between food and water securities in the Arab World in terms of sustainable
development.
**Table 4.** The ratio of saved water and lands to internal water resources and agricultural land area in the Arab World
**Table 5.** Water requirement for increasing 1 % self-sufficiency of study crops in comparison with average self-sufficiency in
the Arab World from 2000 to 2012
**3.3 Analysis of structural changes in virtual water trade network centering the Arab World**
The VWT is regarded as significant element for sustainable water and food management in the Arab World where water is
scarce. Accordingly, in this study we analysed the change of structural connectivity of VWT network in the Arab World using
in-degree centrality from 2000 to 2012, and figured out the vulnerable expansion or reduction in VWT network, which consists
of the volume and number of links. The in-degree centrality based on the number and volume of links in VWT network, which
expressed to non-scaled in-degree centrality (NSInDC) which is based on the number of links, and scaled in-degree centrality
(SInDC) which is based on the volume of links.
Figure 4 showed the NSInDC and SInDC in virtual water trade network by each country in the Arab World in 2012. Egypt
and Yemen showed that NSCInD was lower but SInDC was higher than other countries, and it indicates the intensive
connectivity with a few exporters. In contrast, Saudi Arabia had larger SInDC than other countries expect for Egypt and the
NSCInD was also highest in the Arab World. Accordingly, Saudi Arabia has more distributed structure of VWT. In addition,
UAE and Iraq had similar SInDC in 2012 but NSInDC was quite different; UAE (0.46) and Iraq (0.27). Furthermore, SInDC
in Morocco (96.45) was larger than UAE (83.41) but NSInDC in Morocco (0.26) smaller than UAE (0.46). In comparison to
UAE, Morocco had intensive connection with less exporters than UAE.
Figure 5 showed the temporal changes of NSInDC and the SInDC during two periods (2000–2006 and 2006–2012). In these
results, the Arab World countries were divided into four types (I–IV). Type I countries show a robust expansion in the virtual
water import, and the countries in this type increased the connectivity and volume of virtual water imported, simultaneously.
Type II countries increased the volume of virtual water imported without expansion of connectivity. Type III and type IV
countries show reductions in the virtual water import with and without reduction of connectivity, respectively. In the early
2000s, most of countries in Arab World tried to expand their trade structure by increasing both the connectivity to exporters
and the volume of virtual water imported. In Bahrain, Omen, Qatar, Yemen, Saudi Arabia, Lebanon, and UAE NSInDC of the
VWT network increased significantly from 2000 to 2006, which means that the trade connectivity expanded. The expanded
structure of VWT indicates that the Arab countries is connected to various exporters and it can bring the security of import. In
particular, import of food crops is essential factor in food security in the Arab World, even if they try to increase food self-
sufficiency through increasing domestic production. However. Egypt had the largest SInDC but NSInDC was located 6th in
the Arab World. In 2006, Egypt expanded the connectivity in VWT network, as shown in increasing NSInDC, and Saudi
Arabia also expanded the connectivity.
However, the VWT has become a more vulnerable structure in the Arab World in recent years. Most of the Arab countries
increased the volume of virtual water imported, but the number of exporters that linked to the Arab countries decreased or
increased little from 2006 to 2012. In particular, in 2012 most of countries kept the connectivity or reduced it except for Algeria,
Iraq, Libya, and UAE. For example, virtual water imported in Lebanon significantly increased from 2006 to 2012 but NSInDC
decreased in 2012. Figure 6 showed the change of virtual water import in Lebanon in 2000, 2006, and 2012. In 2000 Lebanon
imported most of virtual water from the USA, Argentina, and Australia, thus VWT in Lebanon was strongly dependent on
these exporters. However, Lebanon expended the VWT in 2006 and Russian federation, Turkey, and Kazakhstan contributed
to virtual water import in Lebanon. Accordingly, the structure of VWT in Lebanon was getting to a distributed network.
However, the VWT in 2012 showed it was dominated by Ukraine and Russian federation even if Lebanon imported more



virtual water in 2012 than 2006. Therefore, Lebanon should consider not only amount of virtual water but also structure of
VWT for sustainable food security in the condition of strong dependency on crop import.
These results indicate that the dependence of the Arab World on virtual water import accelerated recently with the large
increase in volume of virtual water imported. However, the connectivity of the VWT in the Arab World has not increased as
much as the volume of virtual water imported.
**Figure 4.** In-degree centrality of each country in the Arab World in 2012
**Figure 5.** Country types in the Arab World according to the rate of increase in the in-degree centrality from 2000 to 2012
**Figure 6.** Virtual water import from exporters to Lebanon in 2000, 2006, and 2012

We also analyzed the influence of each country on entire VWT network centering the Arab World using eigenvector centrality,
as shown on Figure 7. In 2000, Egypt and Saudi Arabia were identified as the most influential importers in the Arab World
and the USA and Australia were the most influential exporters. Accordingly, the entire VWT in the Arab World could be
affected by these importers and exporters, and it means that the change of trade policy or food management in these countries
could change the structure of VWT in the Arab World. In 2006 and 2012, the influential countries in the Arab World still were
Egypt and Saudi Arabia but the influential exporters moved to Russian federation and Ukraine and Brazil. These results might
contribute to understanding the key player in entire VWT centering the Arab World and other countries in the Arab World
should observe the behavior of influential countries closely.
**Figure 7.** Eigenvector centrality of virtual water trade network in the Arab World at 2000, 2006, and 2012
### 3.4 The importance and limitations of concept of virtual water in the Arab World from a policy perspective
Generally, the VWT is more related to resource management in exporting countries rather than importing countries because
of the embedded water in food trade indicates water resource that is consumed for producing food products in the exporting
country. However, VWT is also considered as an important issue in importing countries in terms of water and food security.
For example, the reduction of VWT might be related to water consumption by replacing imported food products by domestic
food products.
As mentioned above, the VWT can be a major resource in the Arab World. Accordingly, vulnerable VWT, for example low
connectivity, can be a risk element for future food security risk management. In particular, the Arab World is strongly
dependent on food products from exporting countries, and it implies a strong dependency on water resource from exporting
countries. Therefore, water shortages or low food production in exporting countries might cause increasing food price in the
Arab World but also increasing domestic water use for increasing domestic food production.
In this study, we believe that the VWT in the Arab World can be the key factor for bridging water and food, and it is important
to quantify the influence of trade on water and food management. In addition, this study revealed vulnerability (or robust)
expansion (or reduction) and influential trader in VWT network in the Arab World through in-degree and eigenvector centrality
indices. If a country in the Arab World has low connectivity but a large amount of virtual water import, this country should
revaluate their vulnerable trade structure and change the trade policy or water-food management.
However, the application of the concept of VWT is under critical discussion (Wichelns, 2010). First, water footprints bring
new concepts of water management, but it is also difficult to link to operating water resource systems. Water footprint is more
related to water consumption rather than water supply. We can quantify water requirement for producing food products or
water saving by importing them using water footprint and VWT. However, the operation of water facilities, for example
reservoir, desalination plant, and ground water pumping station, are affected by monthly rainfall and ground water level,
development of technology, fertilizer usage, irrigation scheduling and system. Therefore, we need to realize that water footprint
can be changed by various factors. Second, VWT could contribute to connecting water management to food security; however,
food trade is affected by the scarcity or affluence of other important resource such as capital, labor, and land (Biewald et al.,

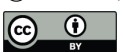



2014). In particular, economic values such as price of food products is the main driver in global food trade but there is no
global value established for virtual water. Therefore, it is difficult to apply virtual water to trade policy in terms of economic
efficiency. Therefore, policy makers or resource manager in the Arab World should consider not only the effects of VWT but
also the difficulty in adapting virtual water to policies for resource management.
Despite these limitations, this study attempted to analyze the VWT through various perspectives. Through the in-degree
centrality of the VWT network, we identified that most countries in the Arab World increased connections with exporters and
the volume of virtual water imported between 2000 and 2006. However, most countries increased the volume of virtual water
imported without increasing the expansion of connections between 2006 and 2012. These results could underscore the fact
that the VWT structure has not recently increased in robustness. We believe that virtual water has a role in achieving
sustainable water, land, and food security, even if there are limitations and difficulties in applying the virtual water concept.

## 4. Conclusions

The VWT, importing water in virtual form, could be a major water portfolio that dominates water management in the water-
scarce countries of the Arab World. Since the virtual water concept was introduced, various studies have been conducted to
quantify the volume of the VWT. In water-deficit areas such as the Arab World, the VWT can offer new perspectives for
understanding and solving water stress and scarcity. The amount of virtual water imported is regarded as the most important
factor in determining water and food security, and the results of water and land savings by crop import in the Arab World
could show the importance of international trade.
In summary, policy makers can benefit by considering both the quantitative impacts of VWT and the structural change of
VWT such as vulnerable expansion (or reduction) in the Arab World. The intensity and connectivity of VWT, which were
analysed in this study, can be major component for integrating food and water policy in the Arab World, and this study might
give important information to policy maker for evaluating future scenarios about resource management toward sustainability
in the Arab World.

## Acknowledgment

We appreciate the use of the national water footprint data from Mekonnen and Hoekstra (2010). The international trade data,
crop production, and harvested arear from 2000 to 2012 are available from the FAOSTAT.

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






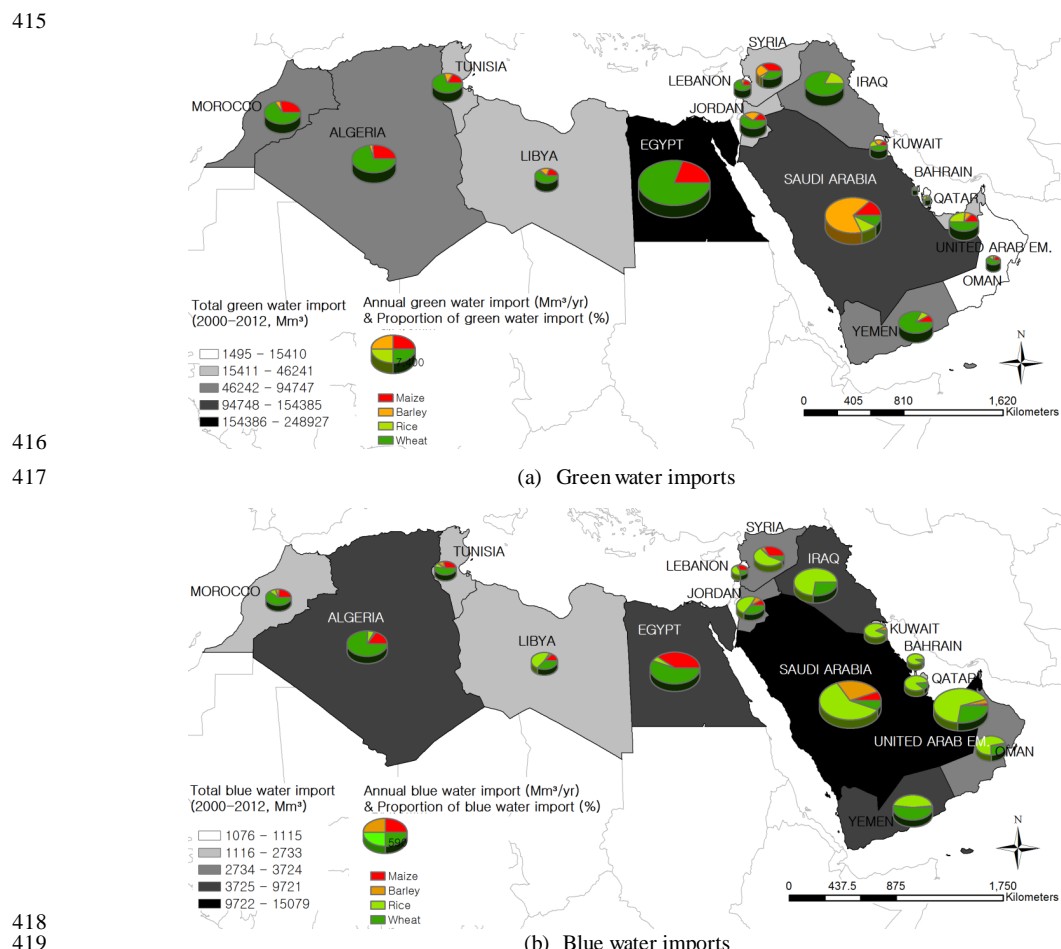

(a)   Green water imports



(b)   Blue water imports

**Figure 1.** The total amount of virtual water imported by each country in the Arab World from 2000 to
2012, separated into green (upper) and blue (lower) water. The pie graph shows the annual import and
proportion of each crop, and the size of the pie indicates the amount of annual virtual water imported
from 2000 to 2012.




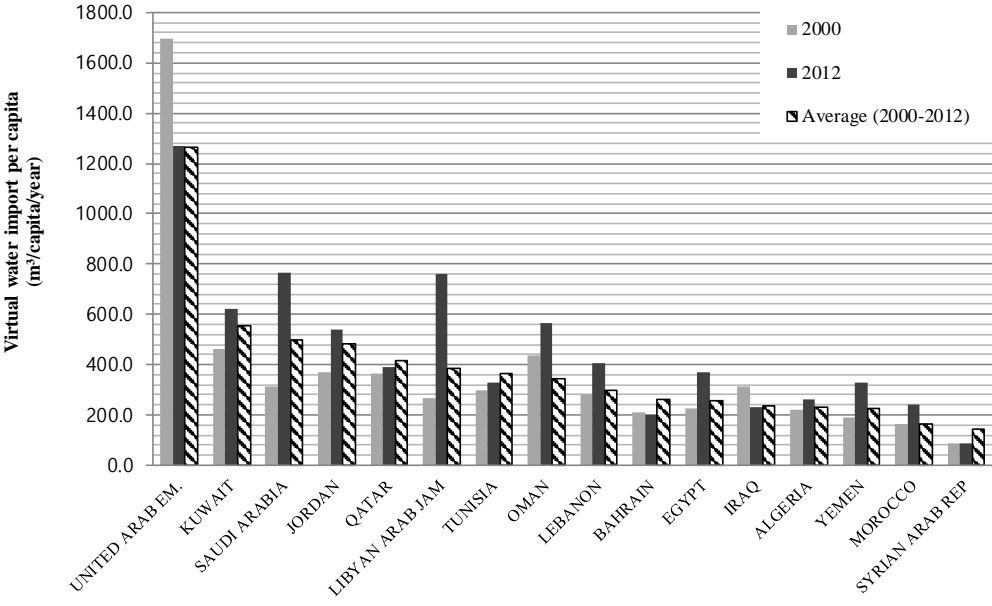


**Figure 2.** Virtual water import per capita in 2000 and 2012.







(a) Barley

(b) Maize

(c) Rice

(d) Wheat

**Figure 3.** The amounts of green water export (GWE) and blue water export (BWE) from the primary
430                     exporters to the Arab World from 2000 to 2012







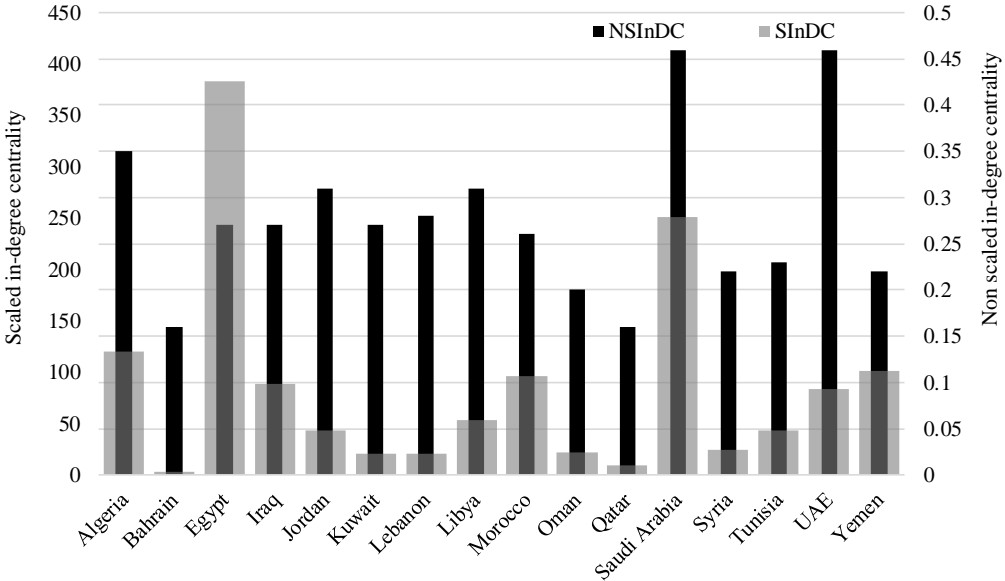

**Figure 4.** In-degree centrality of each country in the Arab World in 2012

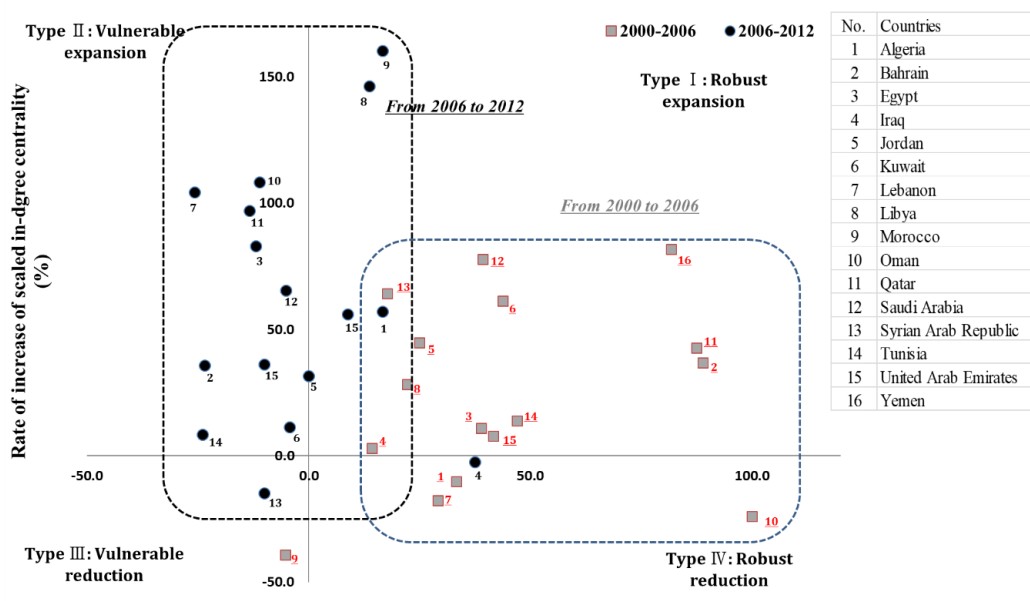

**Figure 5.** Country types in the Arab World according to the rate of increase in the in-degree centrality from 2000 to 2012





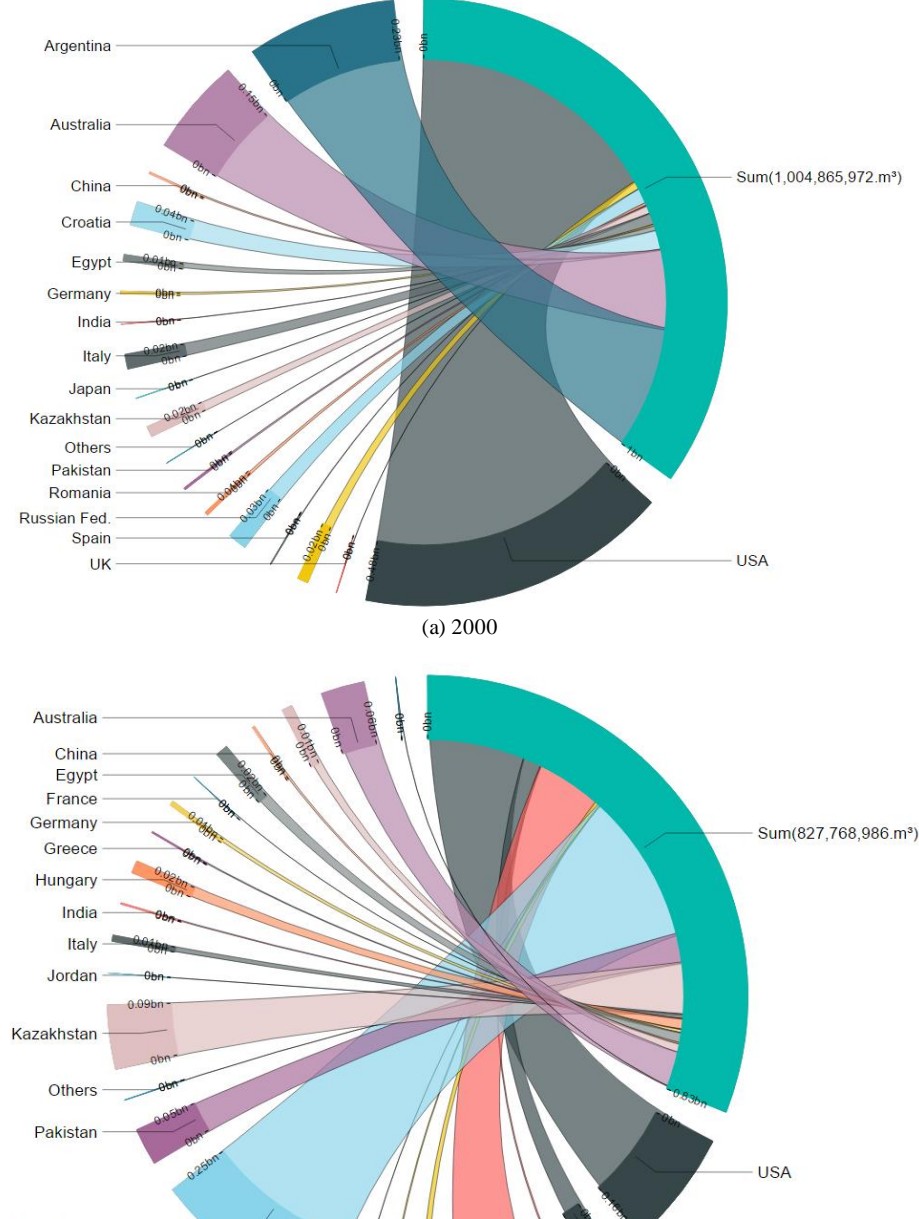


(a) 2000


(b) 2006





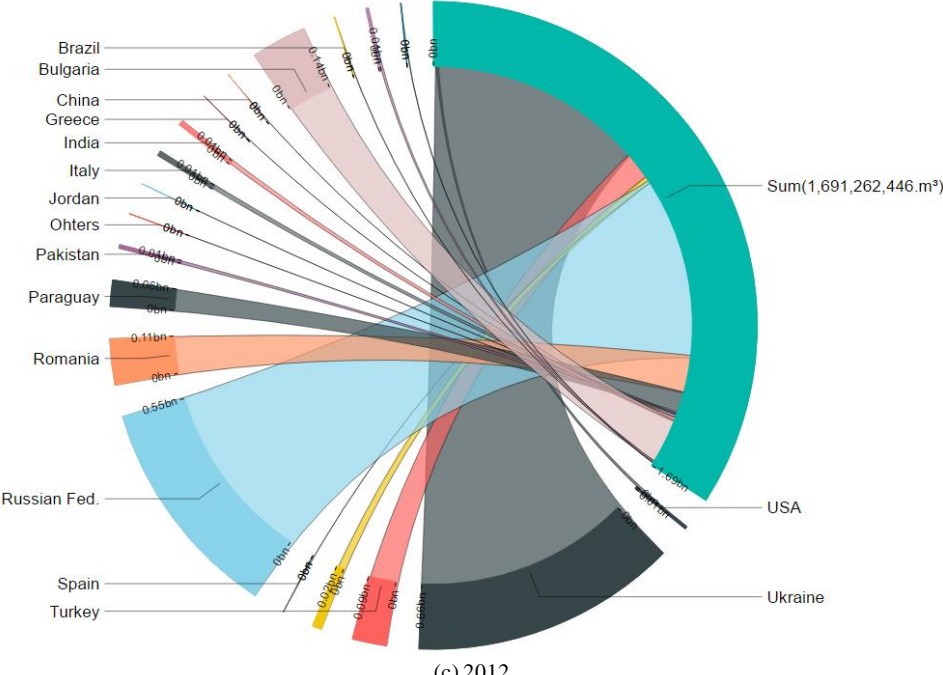

(c) 2012

**Figure 6.** Virtual water import from exporters to Lebanon in 2000, 2006, and 2012

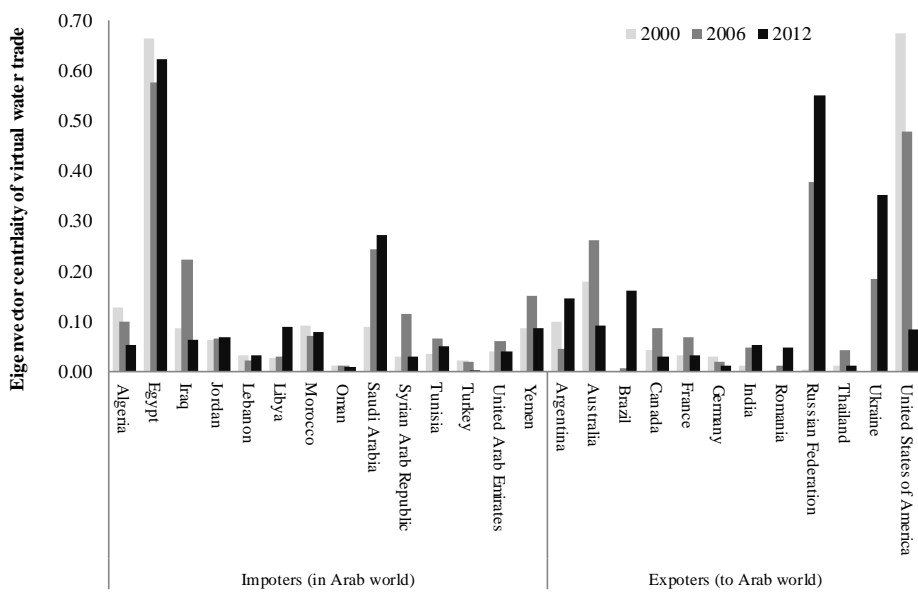

**Figure 7.** Eigenvector centrality of virtual water trade network in the Arab World at 2000, 2006, and 2012





**Table 1.** The amount of crops imported by the Arab World from 2000 to 2012

| Importers in the Arab World | Crop import from 2000 to 2012 | | | | | | | |
|---|---|---|---|---|---|---|---|---|
| | Total import (10⁶ ton) | | | | Annual import (1000 ton/year) | | | |
| | Barley | Maize | Wheat | Rice | Barley | Maize | Wheat | Rice |
| ALGERIA | 3.04 | 27.46 | 69.73 | 0.61 | 234 | 2,113 | 5,364 | 47 |
| BAHRAIN | 0.00 | 0.09 | 0.52 | 0.62 | 0 | 7 | 40 | 48 |
| EGYPT | 0.32 | 65.96 | 107.85 | 0.60 | 25 | 5,074 | 8,296 | 46 |
| IRAQ | 0.25 | 0.23 | 33.10 | 9.65 | 35 | 19 | 2,546 | 742 |
| JORDAN | 6.34 | 5.02 | 10.30 | 1.79 | 488 | 386 | 793 | 137 |
| KUWAIT | 2.32 | 1.75 | 3.70 | 2.23 | 178 | 134 | 285 | 171 |
| LEBANON | 0.64 | 3.77 | 4.78 | 0.60 | 49 | 290 | 367 | 46 |
| LIBYA | 2.94 | 5.58 | 10.45 | 1.59 | 226 | 429 | 804 | 123 |
| MOROCCO | 5.10 | 18.81 | 38.93 | 0.17 | 393 | 1,447 | 2,994 | 13 |
| OMAN | 0.47 | 1.29 | 3.75 | 1.54 | 36 | 100 | 288 | 119 |
| QATAR | 0.43 | 0.05 | 0.62 | 1.14 | 33 | 4 | 48 | 87 |
| SAUDI ARABIA | 81.29 | 20.80 | 9.11 | 13.12 | 6,253 | 1,600 | 701 | 1,009 |
| SYRIA | 5.11 | 17.15 | 5.91 | 2.62 | 393 | 1,319 | 455 | 202 |
| TUNISIA | 5.30 | 9.59 | 19.84 | 0.23 | 407 | 738 | 1,526 | 17 |
| UAE | 2.80 | 5.20 | 13.83 | 8.88 | 215 | 400 | 1,064 | 683 |
| YEMEN | 0.02 | 4.47 | 27.26 | 3.63 | 3 | 344 | 2,097 | 279 |
| **Total** | 116.4 | 187.2 | 359.7 | 49.0 | 8,968 | 14,404 | 27,668 | 3,769 |

Source: FAOSTAT (http://www.fao.org/faostat/)






**Table 2** Cultivation area and production of four major crops in the Arab World.

| Importers in the Arab World | Average cultivation area from 2000 to 2012 (ha/year) | | | |
|---|---|---|---|---|
| | **Barley** | **Maize** | **Wheat** | **Rice** |
| ALGERIA | 760,545 | 308 | 1,658,197 | - |
| EGYPT | 68,103 | 876,153 | 1,180,644 | 625,626 |
| IRAQ | 914,074 | 128,842 | 1,451,219 | 85,182 |
| JORDAN | 31,158 | 947 | 20,116 | - |
| KUWAIT | 1,058 | 290 | 173 | - |
| LEBANON | 13,515 | 949 | 45,380 | - |
| LIBYA | 191,641 | 1,356 | 165,469 | - |
| MOROCCO | 2,118,032 | 226903 | 2,910,977 | 5,876 |
| OMAN | 1,002 | - | 426 | - |
| QATAR | 947 | 94 | 15 | - |
| SAUDI ARABIA | 12,279 | 16,689 | 374,414 | - |
| SYRIA | 1,313,101 | 53,405 | 1,667,229 | - |
| TUNISIA | 385,189 | - | 722,038 | - |
| UAE | 14 | 144 | 18 | - |
| YEMEN | 39,276 | 40,774 | 110,138 | - |
| **Importers in the Arab World** | **Average production from 2000 to 2012 (ton/year)** | | | |
| | **Barley** | **Maize** | **Wheat** | **Rice** |
| ALGERIA | 1,049,710 | 1,128 | 2,313,464 | |
| EGYPT | 134,034 | 6,812,845 | 7,549,253 | 6,023,684 |
| IRAQ | 751,099 | 307,682 | 2,009,972 | 232,040 |
| JORDAN | 22,757 | 17,514 | 23,379 | - |
| KUWAIT | 2,191 | 5,855 | 345 | - |
| LEBANON | 24,834 | 3,579 | 126,623 | - |
| LIBYA | 94,107 | 2,997 | 128,149 | - |
| MOROCCO | 1,867,670 | 159,127 | 4,200,596 | 36,936 |
| OMAN | 3,027 | - | 1,432 | - |
| QATAR | 2,841 | 1,329 | 34 | - |
| SAUDI ARABIA | 68,366 | 86,181 | 1,997,598 | - |
| SYRIA | 817,609 | 211,675 | 4,008,420 | - |
| TUNISIA | 411,431 | - | 1,302,438 | - |
| UAE | 111 | 2,931 | 74 | - |
| YEMEN | 32,248 | 57,329 | 173,437 | - |

Source: FAOSTAT (http://www.fao.org/faostat/)





**Table 3** The amount of virtual water imported by the Arab World from 2000 to 2012.

| Importers in the Arab World | Green water import (10⁶ m³/year) | | | | Blue water import (10⁶ m³/year) | | | |
|---|---|---|---|---|---|---|---|---|
| | **Barley** | **Maize** | **Wheat** | **Rice** | **Barley** | **Maize** | **Wheat** | **Rice** |
| ALGERIA | 242.0 | 1,883.6 | 5,104.8 | 57.8 | 7.8 | 76.6 | 371.1 | 33.5 |
| BAHRAIN | 0.4 | 7.5 | 62.7 | 44.4 | 0.2 | 0.3 | 7.1 | 78.2 |
| EGYPT | 37.3 | 3,798.4 | 15,254.1 | 58.4 | 1.1 | 295.6 | 418.6 | 32.5 |
| IRAQ | 33.2 | 16.7 | 4,645.8 | 1,027.8 | 2.2 | 1.3 | 153.9 | 404.8 |
| JORDAN | 656.8 | 364.2 | 1,483.9 | 81.2 | 20.8 | 20.8 | 84.5 | 115.0 |
| KUWAIT | 257.0 | 159.1 | 557.7 | 211.6 | 9.7 | 2.3 | 10.2 | 138.1 |
| LEBANON | 84.7 | 211.0 | 749.5 | 30.0 | 2.3 | 25.6 | 18.9 | 36.0 |
| LIBYA | 359.6 | 408.9 | 1,245.4 | 56.0 | 8.4 | 26.8 | 75.3 | 99.7 |
| MOROCCO | 318.6 | 1,383.2 | 3,345.0 | 8.9 | 12.1 | 46.1 | 118.8 | 20.4 |
| OMAN | 52.7 | 123.2 | 470.8 | 107.6 | 5.4 | 4.1 | 67.8 | 201.3 |
| QATAR | 50.9 | 6.4 | 76.4 | 77.6 | 2.4 | 0.3 | 19.1 | 146.9 |
| SAUDI ARABIA | 8,154.5 | 1,521.4 | 974.0 | 1,225.9 | 324.3 | 68.9 | 70.8 | 696.0 |
| SYRIA | 556.4 | 947.3 | 900.0 | 120.8 | 12.8 | 90.2 | 17.8 | 165.6 |
| TUNISIA | 409.8 | 611.7 | 2,507.7 | 27.8 | 16.0 | 40.7 | 73.9 | 11.6 |
| UAE | 315.7 | 465.8 | 1,671.8 | 859.5 | 28.5 | 14.3 | 249.3 | 612.5 |
| YEMEN | 3.1 | 406.1 | 3,597.3 | 392.7 | 1.6 | 8.2 | 247.3 | 220.8 |
| **Total** | 11,532.9 | 12,314.5 | 42,646.9 | 4,388.0 | 455.5 | 722.1 | 2,004.4 | 3,012.9 |

**Table 4** The ratio of saved water and lands to internal water resources and agricultural land area in the Arab World

| Importers | Internal water resources* (10⁹ m³) | National blue water saving (10⁹ m³) | Agricultural land* (1000 ha) | National land saving** (1000 ha) |
|---|---|---|---|---|
| ALGERIA | 11.25 | 0.56 | 41432 | 4902 |
| EGYPT | 1.80 | 13.05 | 3761 | 1964 |
| IRAQ | 35.20 | 12.17 | 9230 | 2398 |
| JORDAN | 0.68 | 1.02 | 1057 | 1531 |
| KUWAIT | - | 1.14 | 154 | 229 |
| LEBANON | 4.80 | 0.06 | 658 | 238 |
| LIBYA | 0.70 | 1.73 | 15355 | 1704 |
| MOROCCO | 29.00 | 5.39 | 30401 | 6001 |
| OMAN | 1.40 | 0.69 | 1469 | 100 |
| QATAR | 0.06 | 0.17 | 68 | 32 |
| SAUDI ARABIA | 2.40 | 8.14 | 173295 | 1501 |
| SYRIA | 7.13 | 2.36 | 13921 | 1417 |
| TUNISIA | 4.20 | 0.21 | 9943 | 1288 |
| UAE | 0.15 | 0.82 | 382 | 387 |
| YEMEN | 2.10 | 6.05 | 23546 | 1656 |

* World Bank 2014
** Land saving considered barley, maize, and wheat except for rice because of lack of data.





**Table 5** Water requirement for increasing 1 % self-sufficiency of study crops in comparison with average self-sufficiency in the Arab World from 2000 to 2012

| Importers | Average self-sufficiency from 2000 to 2012 (%) | | | Additional irrigation water requirement ($10^6$ m³) | | |
|---|---|---|---|---|---|---|
| | Barley | Maize | Wheat | Barley | Maize | Wheat |
| ALGERIA | 81.77% | 0.05% | 30.13% | 5.88 | 1.74 | 7.27 |
| EGYPT | 84.28% | 57.31% | 47.64% | 18.31 | 307.44 | 278.77 |
| IRAQ | 95.55% | 94.18% | 44.12% | 983.99 | 122.93 | 233.96 |
| JORDAN | 4.46% | 4.34% | 2.86% | 1.73 | 0.35 | 8.40 |
| KUWAIT | 1.22% | 4.19% | 0.12% | 4.16 | 0.31 | 6.60 |
| LEBANON | 33.63% | 1.22% | 25.65% | 0.00 | 0.04 | 0.65 |
| LIBYA | 29.40% | 0.69% | 13.75% | 8.32 | 0.36 | 16.87 |
| MOROCCO | 82.62% | 9.91% | 58.39% | 10.88 | 57.38 | 43.33 |
| OMAN | 7.76% | 0.00% | 0.49% | 1.00 | 0.08 | 5.70 |
| QATAR | 7.93% | 24.94% | 0.07% | 0.67 | 0.04 | 0.79 |
| SAUDI ARABIA | 1.08% | 5.11% | 74.02% | 51.64 | 22.81 | 118.11 |
| SYRIA | 67.54% | 13.83% | 89.81% | 1.60 | 28.28 | 213.67 |
| TUNISIA | 50.27% | 0.00% | 46.05% | 1.26 | 0.61 | 3.84 |
| UAE | 0.05% | 0.73% | 0.01% | 0.17 | 0.33 | 5.46 |
| YEMEN | 91.49% | 14.28% | 7.64% | - | 13.98 | 58.54 |