# Peer review of "Savings through Food Trade and Structural Changes of Virtual"

_Hydrology and Earth System Sciences, 2018_

## Short Comment (SC1) · 25 Jan 2018

The world population is expected to reach 9.6 billion by 2050, according to a report published by the United Nations(UN). Moreover, despite the progress made over the last two decades by some of the international organizations such as the Food and Agriculture Organization (FAO) of the UN, around 815 million people still suffer from chronic hunger. On top of this, climate change driven by greenhouse gas emissions from human activity and livestock redistributes rainfall patterns, drought, and flooding to threaten our ability to achieve global food security, eradicate poverty, and achieve sustainable development. These issues have alarmed many nations to address the water issues from the perspective of climate change and food security. Therefore, to address the water issues, specifically in water scarce regions, the concept of virtual water trade (VWT, i.e., importing water in virtual form), is actively researched to fuse the concept in water policy formulation.

In this manuscript, using the concept of VWT, the authors analyze the impacts of food trade on food security and water-land savings in the Arab World. Based on this review, the following points are highlighted:

1) The abstract of the manuscript does not state the problem statement. The abstract starts with the aim of the study.

2) In Table 4, how did you compute the national blue water saving. As per the footnote, the values for the national blue water saving are estimated by you. As per your Table 3, if I consider Saudi Arabia, the total blue water imported is (324.3+68.9+70.8+696) * 106 m3/year =1160*106 m3/year=1.160*109 m3/year. However, as per your Table 4, the national blue water saving in Saudi Arabia is 8.14*109 m3/year. You need to provide a sample calculation to support your values. You may also want to see LN 183.

3) As per the authors, the largest amount of blue water was imported annually by Saudi Arabia, followed by the UAE [see LN 183]. This statement contradicts with the values presented in Table 4. As per Table 4, Egypt and IRAQ have saved 13.05*109 m3 and 12.17*109 m3, respectively.

4) The magnitudes/values of virtual water import per capita need to be supported with the population data for the countries. As per the figure, though Egypt and Saudi Arabia import a very large volume of virtual water, UAE has a very high value of virtual water import per capita. What does this lead to conclude? I think, to make a concrete statement, considering the land and water saving, you need to work with the population data distributed by the World Bank's Development Data Group that provides population and other demographic estimates and projections from 1960 to 2050

(https://data.worldbank.org/data-catalog/population-projection-tables). See LN 57-59.

5) In Table 3, what is meant by green water import? Is this the amount of rainfed water [see LN 105-106] in the exporting country? Is this the amount of rainfed water in the importing country? Do we assume that the green water in the importing country is equal to the green water in the exporting country? Does this make sense without considering the climatology/hydrology and other crucial factors in the importing country? I think, few lines (may be from Mekonnen and Hoekstra, 2010) are required from the authors for the readers to understand. As per your equation (1), you are using WEP[ne,c].

6) The equation (3) and equation (4) need to be re-written. Some of the variables are undefined. Moreover, the equations do not have the variable "w".

7) Does the Arab World strongly depend on water resources from exporting countries [see LN 310-311]? I think, based on your Table 4, only some of the countries rely on water resources from exporting countries. In fact, based on the values presented in Table 4, I am unsure the reason for some of the countries (e.g., Algeria) to rely on the exporting countries when they have the capacity. Probably, you need to bring the water price and the local conditions to realize the reason for the import in those countries.

8) The equation that is used to compute the self-sufficiency is not understood. As per the authors, crop import could result in low food self-sufficiency in the Arab World [LN 222]. However, as per the authors, the self-sufficiency is defined as the ratio of imported crops to total consumption [LN 236-237]. Based on this equation, the authors mention in the manuscript that the average self-sufficiency of Wheat in Egypt from 2000 to 2012 was 47.64%. Furthermore, the authors state that 278.77 million m3 irrigation water would be required to increase the self-sufficiency by 1% to reach 48.64% [LN 238-240]. Going by the definition of self-sufficiency, the 1% increase in the self-sufficiency has caused the country (i.e., Egypt) to import an additional 1% from the exporting countries. Does this lead to self-sufficiency?

Minor comments»> see the attached file

[Figure]

Please also note the supplement to this comment:
https://www.hydrol-earth-syst-sci-discuss.net/hess-2018-4/hess-2018-4-SC1-supplement.pdf

**Supplement:**

[revised manuscript text omitted]

---

## Author Comment (AC1) · 13 Feb 2018

Dear Reviewer

We revised the paper in consideration of the reviewer's comments. We tried to answer to the important comments, however, some of the points made require more time to address. Once all of these comments have been addressed, the revised manuscript will be forwarded. We appreciate the feedback and comments, which have contributed to an improved paper.

Please find the answer of your major comments as followed.

1) Comment: The abstract of the manuscript does not state the problem statement. The abstract starts with the aim of the study. A. Answer: I agree. We worked on revising the abstract, and added more statement before mentioning the aim of the study. B. The MENA region (Middle East and North Africa or MENA) has the largest water deficit in the world; it also has the least food self-sufficiency. Increasing food imports while decreasing domestic food production can contribute to water savings and hence to greater water security. However, increased domestic food production is better way to achieve food security, even if irrigation demands increase under projected climate changes. There is trade-off between food security and the savings of water and land through food trade, and this trade-off a significant factor, especially in the MENA. This study analyses the impact of food trade on food security and water-land savings in the MENA region and in terms of virtual water trade (VWT). We estimate the total volume of virtual water imported for four major crops - barley, maize, rice, and wheat – between 2000 and 2012 to assess the impact on water and land savings, and food security. The largest volume of virtual water was imported by Egypt (19.9 billion $m^3$/year), followed by Saudi Arabia (13.0 billion $m^3$/ year). We concluded that Egypt could save 13.1 billion $m^3$ in irrigation water and 2.0 million ha of land area by importing food rather than producing crops. In addition, connectivity and influence of each country in the VWT network was analysed using degree and eigenvector centralities. The study revealed that the MENA region focused more on increasing the volume of virtual water imported during the period 2006-2012 with little attention to the expansion of connections with country exporters: a vulnerable expansion. The study sheds light on opportunities and risks associated with VWT and its role in food security and land management in the MENA region.

2) Comment: In Table 4, how did you compute the national blue water saving. As per the footnote, the values for the national blue water saving are estimated by you. As per your Table 3, if I consider Saudi Arabia, the total blue water imported is

(324.3+68.9+70.8+696) * 106 m3/year =1160*106 m3/year=1.160*109 m3/year. However, as per your Table 4, the national blue water saving in Saudi Arabia is 8.14*109 m3/year. You need to provide a sample calculation to support your values. You may also want to see LN 183. 3) Comment: As per the authors, the largest amount of blue water was imported annually by Saudi Arabia, followed by the UAE [see LN 183]. This statement contradicts with the values presented in Table 4. As per Table 4, Egypt and IRAQ have saved 13.05*109 m3 and 12.17*109 m3, respectively. A. Answer: Blue water import and blue water saving have distinct meanings. The blue water import is calculated by the water footprint of exporters. For example, Saudi Arabia imported wheat from various exporters and blue water import is calculated by multiplying the quantity of imported wheat with the respective blue water footprint of each exporter. However, blue water saving indicates the amount of water to produce the same quantity of imported wheat but as domestic production. Therefore, blue water saving is calculated by multiplying the imported wheat by the blue water footprint of Saudi Arabia. In the revised manuscript, we will include an example of this calculation for clarity. B. Answer: We added more explanation as followed: The import of crops could affect the water and land savings in the importing country. Therefore, the failure of trade could cause water and land shortages in the MENA region. Therefore, we analyzed water and lands requirements for producing as much crop as is imported in each Arab country. Virtual water import and water saving have distinct meanings. The virtual water import is calculated by the water footprint of exporters. For example, Saudi Arabia imported wheat from various exporters and virtual water import is calculated by multiplying the quantity of imported wheat with the respective water footprint of each exporter. In this study, we consider only blue water as resource which can be saved; therefore, water saving indicates the amount of blue water to produce the same quantity of imported wheat but as domestic production.

4) Comment: The magnitudes/values of virtual water import per capita need to be supported with the population data for the countries. As per the figure, though Egypt and Saudi Arabia import a very large volume of virtual water, UAE has a very high

value of virtual water import per capita. What does this lead to conclude? I think, to make a concrete statement, considering the land and water saving, you need to work with the population data distributed by the World Bank's Development Data Group that provides population and other demographic estimates and projections from 1960 to 2050(https://data.worldbank.org/data-catalog/population-projection-tables). See LN 57-59. A. Answer: We considered the amount of virtual water import per capita, which shows the differing viewpoints regarding food and water securities. For example, the population and area of UAE is much smaller than that of Saudi Arabia. Thus, if we consider only total amount of virtual water imported, the UAE might be not considered to be a significant importer. However, the virtual water import per capita in the UAE is larger than that of Saudi Arabia, indicating that the dependency on virtual water imported from exporters in the UAE is much more significant than in Saudi Arabia.

5) Comment: In Table 3, what is meant by green water import? Is this the amount of rainfed water [see LN 105-106] in the exporting country? Is this the amount of rainfed water in the importing country? Do we assume that the green water in the importing country is equal to the green water in the exporting country? Does this make sense without considering the climatology/hydrology and other crucial factors in the importing country? I think, few lines (may be from Mekonnen and Hoekstra, 2010) are required from the authors for the readers to understand. As per your equation (1), you are using WEP[ne,c]. A. Answer: Green water import indicates the green water used in the exporting country to produce crops for export. Actually, there are controversies about the meaning of green water. In this study, we referenced the green water footprint estimated by Makonnnen and Hoekstra (2010). In that study, Green water footprint is water from precipitation that is stored in the root zone of the soil and evaporated, transpired, or incorporated by plants. B. Answer: We added further explanation about the reference of green water footprint (Mekonnen and Hoekstra, 2010).

6) Comment: The equation (3) and equation (4) need to be re-written. Some of the variables are undefined. Moreover, the equations do not have the variable "w". A.

Answer: We removed the sentence "w indicates the water resource such as ground water, surface water, and treated water", and added more explanation about equation. In addition, we added more explanation about equations (3) and (4). B. "where water (or land) saving c,i indicates the amount of water (or lands)to produce the same quantity of imported crop c but as domestic production in importing country i. Import c,i indicate the amount of imported crop c in importing country i. Lands c,i and production c,i indicate the average cultivated area and production of crop c in importing country i.

7) Comment: Does the Arab World strongly depend on water resources from exporting countries [see LN 310-311]? I think, based on your Table 4, only some of the countries rely on water resources from exporting countries. In fact, based on the values presented in Table 4, I am unsure the reason for some of the countries (e.g., Algeria) to rely on the exporting countries when they have the capacity. Probably, you need to bring the water price and the local conditions to realize the reason for the import in those countries. A. Answer: It may be related to various situations that are hardly defined. We are working on explaining this and will add description in the revised manuscript.

8) Comment: The equation that is used to compute the self-sufficiency is not understood. As per the authors, crop import could result in low food self-sufficiency in the Arab World [LN 222]. However, as per the authors, the self-sufficiency is defined as the ratio of imported crops to total consumption [LN 236-237]. A. Answer: Self-sufficiency is defined as the ratio of domestic production to total consumption. Therefore, the ratio of imported crops to total consumption is related to negative self-sufficiency. B. We added paragraph about food self-sufficiency as followed: We applied the concept of self-sufficiency as the index of food security, which is defined as the ratio of domestic production to total consumption, and estimated water requirement of increasing 1 % self-sufficiency of study crops in comparison to average self-sufficiency from 2000 to 2012. In order to increase self-sufficiency of crop, the increase of domestic production should be accompanied, and it derives additional water and land requirement which

can be issue of trade-offs between food security and water-land savings.

9) Comment: Based on this equation, the authors mention in the manuscript that the average self-sufficiency of Wheat in Egypt from 2000 to 2012 was 47.64%. Furthermore, the authors state that 278.77 million m3 irrigation water would be required to increase the self-sufficiency by 1% to reach 48.64% [LN 238-240]. Going by the definition of self-sufficiency, the 1% increase in the self-sufficiency has caused the country (i.e., Egypt) to import an additional 1% from the exporting countries. Does this lead to self-sufficiency? A. Answer: Self-sufficiency indicates the ratio of domestic production to total consumption. Therefore, the increase of 1% self-sufficiency drives the requirement of domestic production.

Please also note the supplement to this comment:
https://www.hydrol-earth-syst-sci-discuss.net/hess-2018-4/hess-2018-4-AC1-supplement.pdf

**Supplement:**

**Table 1 Crops import (ton/year)**

| Arab world | 2000 | 2001 | 2002 | 2003 | 2004 | 2005 | 2006 | 2007 | 2008 | 2009 | 2010 | 2011 | 2012 |
|---|---|---|---|---|---|---|---|---|---|---|---|---|---|
| ALGERIA | 6,646,120 | 5,693,894 | 8,067,136 | 5,536,267 | 6,703,246 | 8,211,321 | 7,056,677 | 7,264,381 | 9,417,266 | 7,481,183 | 7,851,212 | 11,437,793 | 9,475,459 |
| BAHRAIN | 57,707 | 102,490 | 52,253 | 87,534 | 69,013 | 72,347 | 84,473 | 69,169 | 132,018 | 76,654 | 146,875 | 157,891 | 119,903 |
| EGYPT | 11,229,288 | 11,437,570 | 12,212,886 | 10,899,718 | 12,651,637 | 12,688,572 | 12,131,985 | 13,120,439 | 12,357,425 | 14,544,743 | 16,802,187 | 16,620,186 | 18,034,606 |
| IRAQ | 3,869,614 | 3,358,493 | 2,798,285 | 1,311,527 | 3,153,530 | 3,425,835 | 4,065,189 | 3,211,623 | 3,721,860 | 3,842,655 | 2,735,965 | 3,917,074 | 3,811,587 |
| JORDAN | 1,490,142 | 1,526,313 | 1,396,001 | 1,508,025 | 1,763,075 | 2,065,236 | 2,215,151 | 2,126,330 | 1,660,731 | 1,702,436 | 1,700,718 | 2,164,266 | 2,126,812 |
| KUWAIT | 620,910 | 631,337 | 592,716 | 571,835 | 658,087 | 822,760 | 826,412 | 752,023 | 883,851 | 971,835 | 822,566 | 928,773 | 913,123 |
| LEBANON | 739,605 | 642,507 | 636,732 | 582,780 | 888,737 | 768,438 | 544,398 | 660,770 | 585,014 | 903,358 | 829,174 | 960,572 | 1,039,668 |
| Libya | 1,056,456 | 650,894 | 1,046,820 | 741,755 | 1,213,379 | 1,354,253 | 1,441,031 | 1,562,395 | 1,621,199 | 2,859,792 | 2,930,410 | 1,507,342 | 2,578,311 |
| MOROCCO | 5,415,025 | 4,260,818 | 4,657,833 | 3,389,137 | 3,814,163 | 4,991,302 | 3,327,484 | 5,922,861 | 5,757,331 | 4,240,280 | 5,601,939 | 5,623,514 | 6,012,268 |
| OMAN | 592,460 | 463,848 | 304,879 | 414,784 | 375,981 | 405,121 | 506,072 | 525,966 | 763,691 | 395,250 | 698,921 | 684,114 | 918,671 |
| QATAR | 117,400 | 109,530 | 73,597 | 84,927 | 93,636 | 117,272 | 174,391 | 175,300 | 238,414 | 197,619 | 200,472 | 284,001 | 373,466 |
| SAUDI ARABIA | 8,017,852 | 5,874,237 | 8,145,896 | 9,926,192 | 6,682,593 | 9,113,213 | 8,645,945 | 8,979,964 | 10,702,809 | 11,497,013 | 12,121,694 | 11,026,569 | 13,585,818 |
| Syrian Arab Republic | 1,542,296 | 1,049,781 | 1,843,163 | 1,415,923 | 1,679,547 | 2,897,628 | 2,436,884 | 2,124,291 | 3,609,734 | 5,102,701 | 3,195,789 | 2,545,687 | 1,354,659 |
| TUNISIA | 2,763,859 | 2,274,435 | 3,550,933 | 2,193,273 | 1,927,967 | 2,331,940 | 2,793,533 | 3,727,697 | 3,361,108 | 1,984,769 | 3,026,670 | 2,629,120 | 2,385,318 |
| United Arab Emirates | 2,365,130 | 2,046,601 | 2,120,484 | 2,181,351 | 2,140,137 | 1,747,471 | 2,233,009 | 2,210,927 | 2,349,127 | 2,575,125 | 2,727,957 | 2,411,592 | 3,605,413 |
| YEMEN | 1,975,376 | 1,853,123 | 1,922,545 | 2,059,170 | 1,852,955 | 2,501,882 | 3,191,604 | 3,001,187 | 2,202,418 | 3,958,262 | 3,382,608 | 3,252,614 | 4,229,346 |
| Total | 48,499,240 | 41,975,871 | 49,422,159 | 42,904,198 | 45,667,683 | 53,514,591 | 51,674,238 | 55,435,323 | 59,363,996 | 62,333,675 | 64,775,157 | 66,151,108 | 70,564,428 |

**Table 2 Green water import (million m³/year)**

| Arab world | 2000 | 2001 | 2002 | 2003 | 2004 | 2005 | 2006 | 2007 | 2008 | 2009 | 2010 | 2011 | 2012 |
|---|---|---|---|---|---|---|---|---|---|---|---|---|---|
| ALGERIA | 6,325 | 5,191 | 9,717 | 5,357 | 6,768 | 8,183 | 5,846 | 6,695 | 8,116 | 6,176 | 6,803 | 10,194 | 9,376 |
| BAHRAIN | 77 | 109 | 47 | 86 | 79 | 75 | 99 | 78 | 162 | 85 | 210 | 231 | 159 |
| EGYPT | 14,567 | 14,033 | 15,167 | 13,115 | 17,840 | 17,229 | 16,157 | 20,399 | 19,977 | 20,700 | 23,569 | 26,014 | 30,160 |
| IRAQ | 6,851 | 6,068 | 5,347 | 2,379 | 5,701 | 6,010 | 6,399 | 4,970 | 6,644 | 5,796 | 4,467 | 7,081 | 6,475 |
| JORDAN | 1,730 | 2,099 | 2,520 | 2,921 | 2,905 | 2,280 | 2,238 | 2,830 | 2,859 | 2,725 | 2,145 | 3,090 | 3,279 |
| KUWAIT | 785 | 949 | 903 | 853 | 1,047 | 1,261 | 1,286 | 1,077 | 1,260 | 1,654 | 1,292 | 1,558 | 1,485 |
| LEBANON | 934 | 803 | 955 | 909 | 1,213 | 1,174 | 735 | 984 | 796 | 1,356 | 1,155 | 1,366 | 1,598 |
| Libya | 1,213 | 707 | 1,384 | 1,037 | 1,457 | 1,569 | 1,523 | 2,236 | 2,023 | 4,263 | 3,614 | 1,985 | 3,898 |
| MOROCCO | 4,684 | 4,130 | 5,913 | 3,492 | 4,484 | 5,074 | 2,822 | 6,233 | 5,299 | 4,266 | 5,804 | 5,787 | 7,737 |
| OMAN | 889 | 586 | 360 | 421 | 515 | 569 | 610 | 903 | 1,032 | 569 | 1,050 | 1,053 | 1,249 |
| QATAR | 153 | 132 | 67 | 90 | 90 | 150 | 247 | 241 | 323 | 238 | 223 | 312 | 482 |
| SAUDI ARABIA | 5,760 | 5,734 | 10,505 | 11,644 | 8,888 | 10,564 | 11,336 | 10,366 | 13,968 | 16,798 | 15,185 | 14,593 | 19,044 |
| Syrian Arab Republic | 1,197 | 667 | 2,220 | 1,329 | 1,246 | 2,866 | 1,992 | 1,339 | 4,681 | 7,159 | 3,502 | 2,856 | 1,763 |
| TUNISIA | 2,625 | 2,247 | 5,277 | 2,632 | 2,348 | 2,954 | 3,057 | 6,474 | 5,453 | 2,730 | 3,523 | 3,513 | 3,409 |
| United Arab Emirates | 3,183 | 2,428 | 2,468 | 2,306 | 2,830 | 2,564 | 3,073 | 3,114 | 3,362 | 3,506 | 4,297 | 4,021 | 5,915 |
| YEMEN | 3,016 | 2,811 | 2,750 | 2,531 | 3,094 | 4,537 | 5,651 | 5,534 | 3,529 | 5,852 | 5,113 | 5,352 | 7,404 |
| Total | 53,988 | 48,694 | 65,600 | 51,100 | 60,505 | 67,059 | 63,074 | 73,472 | 79,483 | 83,873 | 81,950 | 89,007 | 103,432 |

**Table 3 Blue water import (million m³/year)**

| Arab world | 2000 | 2001 | 2002 | 2003 | 2004 | 2005 | 2006 | 2007 | 2008 | 2009 | 2010 | 2011 | 2012 |
|---|---|---|---|---|---|---|---|---|---|---|---|---|---|
| ALGERIA | 569 | 597 | 798 | 444 | 311 | 515 | 345 | 474 | 844 | 553 | 150 | 410 | 347 |
| BAHRAIN | 57 | 93 | 68 | 105 | 72 | 76 | 84 | 95 | 105 | 81 | 105 | 86 | 89 |
| EGYPT | 718 | 728 | 586 | 803 | 681 | 630 | 792 | 818 | 609 | 952 | 718 | 829 | 857 |
| IRAQ | 281 | 436 | 446 | 306 | 462 | 608 | 943 | 704 | 574 | 547 | 669 | 631 | 686 |
| JORDAN | 114 | 138 | 133 | 171 | 236 | 347 | 431 | 422 | 165 | 220 | 246 | 276 | 235 |
| KUWAIT | 130 | 127 | 132 | 115 | 126 | 167 | 189 | 206 | 224 | 158 | 163 | 188 | 158 |
| LEBANON | 71 | 91 | 62 | 60 | 107 | 92 | 93 | 62 | 51 | 131 | 69 | 93 | 94 |
| Libya | 142 | 99 | 153 | 112 | 145 | 240 | 214 | 216 | 163 | 296 | 341 | 227 | 383 |
| MOROCCO | 257 | 373 | 317 | 160 | 117 | 153 | 179 | 309 | 227 | 141 | 142 | 116 | 75 |
| OMAN | 217 | 220 | 168 | 244 | 206 | 172 | 232 | 195 | 486 | 240 | 336 | 398 | 505 |
| QATAR | 117 | 111 | 98 | 90 | 126 | 107 | 139 | 148 | 192 | 211 | 264 | 314 | 277 |
| SAUDI ARABIA | 1,166 | 942 | 906 | 1,060 | 1,006 | 1,130 | 973 | 1,355 | 1,222 | 1,315 | 1,366 | 1,308 | 1,330 |
| Syrian Arab Republic | 231 | 208 | 203 | 221 | 280 | 370 | 353 | 336 | 202 | 547 | 320 | 217 | 237 |
| TUNISIA | 219 | 170 | 107 | 72 | 90 | 87 | 179 | 272 | 155 | 109 | 161 | 128 | 100 |
| United Arab Emirates | 835 | 1,322 | 1,205 | 1,323 | 1,271 | 626 | 1,254 | 694 | 491 | 546 | 788 | 563 | 841 |
| YEMEN | 295 | 444 | 358 | 655 | 398 | 309 | 363 | 519 | 429 | 500 | 443 | 703 | 789 |
| Total | 5,419 | 6,101 | 5,742 | 5,941 | 5,634 | 5,628 | 6,764 | 6,824 | 6,138 | 6,548 | 6,280 | 6,487 | 7,004 |

---

## Referee Comment (RC1) · Anonymous Referee #1 · 15 Feb 2018

The aim of the manuscript is to "analyze the quantitative and structural characteristics of virtual water trade in the Arab World in order to understand the effects on water savings and land tenure from importing crops and identify the temporal changes of VWT". The authors develop an exhaustive study of virtual water trade of the Arab world, giving figures of water footprints, main exporters, main importers, VWT, water savings and "land savings" for each country. I must say I was very interested in the beginning particularly because of the water scarcity encountered in the Arab world. Although the results are interesting and may be used by stake holders and authorities, the method-

ology is not properly described, and the quality of the figures and introduction hampers the understanding and reproducibility of the paper.

I have a feeling that "water savings" and VWT is the same. Please confirm. If so, why do you have two different equations.

The analysis on land savings has so limited information that it basically needs to be inferred from what is available.

Inconsistencies found in the introduction, methods and visualization of results are mentioned below. The manuscript still needs considerable polishing. There is a very poor explanation on the main equations used (Eq. 3 and 4) and understanding the methods section of centrality is almost impossible. Figure explanations are very limited, with captions of one line.

Detailed review:

L. 22-24 First sentence is convoluted. Please divide in two parts so that it is clearer.

L. 32-33 Redundant. Water trade will contribute. . . in the event of an increase in global food trade. Sentence needs improvement.

L. 41 with the "real" water you mean blue water, green water or both combined?

L. 42-43 A citation for this sentence?

L. 45 "were" saved

L. 47-48 Strange, how can crop trade save virtual water. Isn't virtual water embedded in crop trade? Or do you mean in the importing country? Is this sustainable?

L. 50 25-75 km3 "per year"?

L. 51 What do you mean with "blue water saving from international trade"

L. 51 Again, where is the blue water being saved by the food trade? Be more specific. In total, in the exporting country or in the importing country, or in both?

[Figure]

L. 65 Are you sure of this? You, say later that rain will decrease.

L. 71-72 I think this sentence is repeated from the beginning.

L. 77 Main objective. You have not said what are these structural characteristics of virtual water.

L.. 83 It is not possible to understand this sentence. "increasing 1 % self-sufficiency of study crops in comparison to average self-sufficiency from 2000 to 2012 in terms of trade-off between water saving and food self-sufficiency."

L. 87 What do you mean with "vulnerable expansion" and "robust expansion"?

L. 99 "VWT denotes the VWT..." ??

L. 99 ", ne,"??

L. 111 "The import of crops could affect the water and land savings in the importing country". This is contradictory. So, finally, is virtual trade causing or preventing water and land shortages?

Equation 3 What does "import" mean in Equations 3 and 4? It is not clear. Why isn't the water saving just the VWT? And blue only, or blue and green combined? And what if the crops imported use green water? Or there is an error with this important equation because water times a volume of import cannot give water again, unless I understand incorrectly.

L. 119 I cannot see the "w" in the equations. And what does the variable "Lands" and "land savings" mean?

Section 2.3 was impossible to understand and the level of English has dropped considerably. Please rewrite this section. An explanatory figure of degree centrality would help considerably in understanding this section. Example: "The in-degree centrality based on the number and volume of links in VWT network, which expressed to non-scaled in-degree centrality (NSInDC) which is based on the number of links, and scaled

in-degree centrality (SInDC) which is based on the volume of links."???

L 181 Is this the total volume of water per year or the total volume in the period 2002-2012?

L. 186 Reference needed for "Rice is a blue-water-intensive crop"

Figure 6- What are the small numbers in the figures. Any units? Why is the shape of the flows changing direction? Is that important?

Table 4- How can the water saving be larger than the available water resources, for instance, for Egypt or Saudi Arabia? Is then water saving=virtual water flow?

L256-262 But why is this figure important. You describe it in detail but do not say what this finding represent in real words.

Improve the caption of Figure 4. It needs more explanation.

Improve the caption of Figure 5. It needs more explanation. What do red numbers mean?

---

## Referee Comment (RC2) · Anonymous Referee #2 · 9 Mar 2018

Review for HESS doi.org/10.5194/hess-2018-4
**Analysis of Trade-offs between Food Security and Water-Land Savings through Food Trade and Structural Changes of Virtual Water Trade in the Arab World**

**Summary:**

The authors describe an effort to quantify the virtual trade of water into and out of the Arab countries, as well as the water acquirements associated with increasing self-sufficiency (decreasing import versus increasing domestic food production). They also analyzed the connectivity of virtual trade of water related to the Arab World. They argue that this information can be used for informing policy related to the nexus of water management and agricultural production.

**Recommendation**: Major revisions

**Major Comment:**

Overall, this is an interesting article with some interesting finding. My major comments are primarily related to the following:

- Usability of knowledge gained: The authors could go a little bit further in describing how this information is useful to policy or management. See specific comments 2 and 32.
- Ignoring of significant limitations: While the authors do discuss some of the limitations, there are some very important limitations that are entirely ignored. These should be discussed as well as the potential for these limitations to bias (and in which direction) the results/conclusions. The limitation I am most concerned with is related to specific comment 24, but there are other as well, as detailed below.
- The writing is in general rather different to comprehend. I make multiple suggestions below on how it can be improved and where the confusions lie – but those are really only a start. Furthermore, there are numerous grammatical/typological errors that need correcting. Again only a few are mentioned below.
- Finally, the conclusions are poorly written and not fully developed – it is not clear exactly what was learned that would further policy development/analysis or improved water/food management.

**Specific Comments:**

1. Abstract: the abstract is somewhat impenetrable. It would be nice to have a bit more explanation on some of the concepts/jargon used.
2. Introduction: So the introduction provides a strong argument for the need to look at VWT, but then (after the objective is introduced) provides limited rationale for how this particular effort then feeds back into how it could be used for policy analysis/development. What is the rationale for this particular study within context of the broader VWT area of research?
3. L31-32: To what extent was the energy sector (and impacts to the environment, such as increased greenhouse gas concentrations) considered in the recommendation that water stress could be relieved through increased global food trade? I.e., there are negative consequences when considering this recommendation from the perspective of the food-energy-water(-environment) nexus. This should be mentioned somewhere.
4. L35: It would be helpful when giving this background information to include mechanisms. What is the reason for the ag water deficit? To support growth in population and need for food? Or is this for existing irrigated lands and the fact that climate change is causing increased vulnerability of these water rights in some locations?

5.  L39-43: This is confusing. Is VWT quantified as the water that would have been needed to grow the food if it were grown locally? Otherwise, I don't understand how there can be an imbalance between VWT and "Real Water". BTW, "Real Water" should be defined. It might be nice to have a table of definitions somewhere: VWT, real water, blue water, green water, etc.
6.  L51: blue water has not been defined, nor its relationship to VWT described.
7.  L82-83: Why is this a particularly critical issue in the Arab world?
8.  L83-84: This is a little bit confusing, possibly because the methods have not been presented yet, so it is not clear at this point what is meant by increasing 1% of self-sufficiency.
9.  L86: Is there a citation that can go with "degree centrality" to insert here for those looking for background on it? Same with "eigenvector centrality". It would be nice if there would be a brief explanation of what these are, esp. given the broad readership of this journal.
10. L90 - "is as comparison as". Also, please provide an explanation for this statement - why these two tasks are both critically important, rather than just referring readers to recent literature for the explanation.
11. L99 - there are problems with the indices of the variables in the text (i.e. no subscript). It is also helpful to italicize the variables so it is clear that they are variables.
12. L101 - so this goes back to my earlier confusion on if VWT is determined based on the WFP of the importing or exporting country. Here, it looks like it is calculated on the exporting country (is this true for regardless if it is import VWT or export VWT?), in which case I am still confused on why there is a not a net zero of VWT at the global scale (i.e., import VWT - export VWT = 0 if WFP is determined always in the place where it is grown versus the place it would have been grow if there was no import). Certainly at the scale of a region or country, this would not need to be a net zero (because there could be, e.g. more imports than exports), but it is unclear in the context of where that was described if it is regional or global. This is likely to confuse several readers and so should be more clearly explained in the introduction.
13. L117 - why is only blue water considered in the water saving quantification? Green water should also be considered because most of the irrigation applied to crops is consumptively used, so lost to the system. This results in less water available for other uses. To be most robust, however, the water saving should be based on just the consumptive use portion of the sum of green and blue water.
14. L125 - define what is meant by "edge" versus "node" in the context of this study
15. L126 - explanation should be provided for what is meant by in-degree versus out-degree. i.e. "depending on the direction of trade (in = import; out = export)."
16. L132: Why (N-1)? is this like the Bessel correction for standard deviation?
17. L144-146: Not all of the variables in equations 6 and 7 are defined in the text.
18. L162: Although other data sources were used to estimate the water and land footprints of food production, there should still be a discussion of the limitations to the utilized datasets/approaches in terms of what these limitations mean for the conclusions derived from this study.
19. L173-174: awkward phrasing: "and the part of periods for water footprint is overlapped with the period of trade data"
20. L174-175: It is good to mention limitations, but it is even more useful to mention how these limitations may bias results/conclusions.
21. L194-195 - because of a relative low population in the UAE relative to the other Arab countries? Might be helpful to state that.
22. L197-200 - there are grammatical issues with this sentence.
23. L222 - this sentence also has grammatical issues.
24. L228 - not all ag lands are appropriate for growing every type of crop - this is going to be important to mention as a limitation. Also, some ag lands may produce very high yields of a particular crop than other land area in other country, even if receiving full water requirement. The role of temperature and soil characteristics play a strong role here and this should be mentioned in the discussion and in limitations. Therefore, this is not an apple-to-apple comparison in terms of

the amount of food produced in each region given the water and land needs. This will be important to consider if policies are developed or analyzed given this information.

25. Table 5 - wouldn't increasing self sufficiency also create a hit on land requirement as well? Another limitation related to increasing self sufficiency is that there is no analysis done on whether or not arable land and water are available to do this. So I'm not sure how this can be useful for policy analysis without at least describing these constraints as potentially major limitations.

26. L254-255 - nowhere in the methods is there discussion on how the InDC metrics are scaled. It is not clear from the current text what this is or how it was done (or even why). Therefore, I find it difficult to comment on the results from this analysis.

27. Figure 5 - it is not clear what the black numbers are versus the red numbers. Also, the country numbers are given - do those refer to the red numbers? Why don't the black numbers have country numbers - or do they? except that some of the numbers are very large (e.g., 50, 100) - or is that the y-axis? It's very confusing...

28. L271 - what is meant by "bring the security of import"?

29. Figure 6 - what are the units of these? It's hard to see the light text on each line - what do they say?

30. L300-301 - this is a very awkward sentence.

31. L304-306 - this sentence is important and relates back to the fact that the water/land footprints in the exporting country are very unlikely to match those (for the same food produced) in an importing country. I'd say this is the most important limitation to this study and should be discussed with respect to how this is likely to bias the results (esp. given that Arab countries are relatively hot and yields often are smaller in hot climates and require more water).

32. L330 - but HOW should policy makers do this? What is this missing step in translating the knowledge gained from this study towards analyzing/developing policy (without becoming policy prescriptive of course)?

33. Figure 7: "Impoters" and "Expoters" are misspelled in the figure.

34. Table 4: The caption says that these values are ratios but the values have units - so are they ratios or are they the absolute values of the water/land saved?

35. Table 5: Caption needs to reflect that there are values given for both percent change and magnitude of change.....

---

## Author Comment (AC2) · 7 Apr 2018

Dear Editor and Reviewer

We revised the previous paper with reflection of reviewer's comments. Most of the comments were related to the limitation of this study, and more explanation for clarifying the methodology and results. We tried to incorporate our responses to these comments within the revised manuscript. Please find the revision note and we explained the detailed change in it. We appreciate the feedback and comments and

we believe that these comments improved this paper.

Please also note the supplement to this comment:
https://www.hydrol-earth-syst-sci-discuss.net/hess-2018-4/hess-2018-4-AC2-
supplement.zip

―――――――――――――――――